# Learning Multi-Level Hierarchies with Hindsight

**Andrew Levy**
Department of Computer Science
Brown University
Providence, RI, USA
`andrew_levy2@brown.edu`

**George Konidaris**
Department of Computer Science
Brown University
Providence, RI, USA
`gdk@cs.brown.edu`

**Robert Platt**
College of Computer and Information Science
Northeastern University
Boston, MA, USA
`rplatt@ccs.neu.edu`

**Kate Saenko**
Department of Computer Science
Boston University
Boston, MA, USA
`saenko@bu.edu`

## Abstract

Hierarchical agents have the potential to solve sequential decision making tasks with greater sample efficiency than their non-hierarchical counterparts because hierarchical agents can break down tasks into sets of subtasks that only require short sequences of decisions. In order to realize this potential of faster learning, hierarchical agents need to be able to learn their multiple levels of policies in parallel so these simpler subproblems can be solved simultaneously. Yet, learning multiple levels of policies in parallel is hard because it is inherently unstable: changes in a policy at one level of the hierarchy may cause changes in the transition and reward functions at higher levels in the hierarchy, making it difficult to jointly learn multiple levels of policies. In this paper, we introduce a new Hierarchical Reinforcement Learning (HRL) framework, *Hierarchical Actor-Critic (HAC)*, that can overcome the instability issues that arise when agents try to jointly learn multiple levels of policies. The main idea behind HAC is to train each level of the hierarchy independently of the lower levels by training each level as if the lower level policies are already optimal. We demonstrate experimentally in both grid world and simulated robotics domains that our approach can significantly accelerate learning relative to other non-hierarchical and hierarchical methods. Indeed, our framework is the first to successfully learn 3-level hierarchies in parallel in tasks with continuous state and action spaces. We also present a video of our results and software to implement our framework.

## 1 Introduction

Hierarchy has the potential to accelerate learning in sequential decision making tasks because hierarchical agents can decompose problems into smaller subproblems. In order to take advantage of these shorter horizon subproblems and realize the potential of HRL, an HRL algorithm must be able to learn the multiple levels within the hierarchy in parallel. That is, at the same time one level in the hierarchy is learning the sequence of subtasks needed to solve a task, the level below should be learning the sequence of shorter time scale actions needed to solve each subtask. Yet the existing HRL algorithms that are capable of automatically learning hierarchies in continuous domains (Schmidhuber, 1991; Konidaris & Barto, 2009; Bacon et al., 2017; Vezhnevets et al., 2017; Nachum et al., 2018) do not efficiently learn the multiple levels within the hierarchy in parallel. Instead, these algorithms often resort to learning the hierarchy one level at a time in a bottom-up fashion.

Learning multiple levels of policies in parallel is challenging due to non-stationary state transition functions. In nested, multi-level hierarchies, the transition function for any level above the ground level depends on the current policies below that level. For instance, in a 2-level hierarchy, the

$\Pi_i : \text{State, Goal State} \rightarrow \text{Action}$

$\Pi_2 : \theta, \dot{\theta},$    $\rightarrow$

$\Pi_1 : \theta, \dot{\theta},$    $\rightarrow$

$\Pi_0 : \theta, \dot{\theta},$    $\rightarrow$   Joint Torques

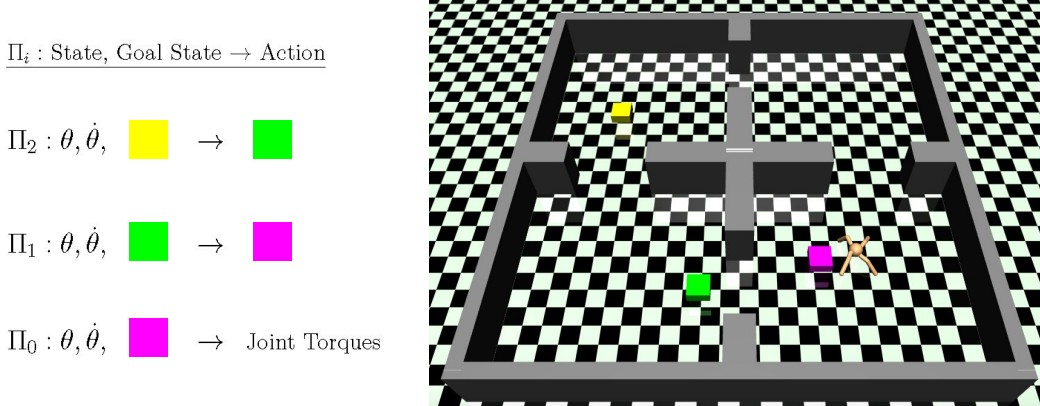

Figure 1: An ant agent uses a 3-level hierarchy to traverse though rooms to reach its goal, represented by the yellow cube. $\Pi_2$ uses as input the current state (joint positions $\theta$ and velocities $\dot{\theta}$) and goal state (yellow box) and outputs a subgoal state (green box) for $\Pi_1$ to achieve. $\Pi_1$ takes in the current state and its goal state (green box) and outputs a subgoal state (purple box) for $\Pi_0$ to achieve. $\Pi_0$ takes in the current state and goal state (purple box) and outputs a vector of joint torques.

high-level policy may output a subgoal state for the low level to achieve, and the state to which this subgoal state leads will depend on the current low-level policy. When all policies within the hierarchy are trained simultaneously, the transition function at each level above ground level will continue to change as long as the policies below that level continue to be updated. In this setting of non-stationary transition functions, RL will likely struggle to learn the above ground level policies in the hierarchy because in order for RL methods to effectively value actions, the distribution of states to which those actions lead should be stable. However, learning multiple policies in parallel is still possible because the transition function for each level above ground level will stabilize once all lower level policies have converged to optimal or near optimal policies. Thus, RL can be used to learn all policies in parallel if each level above ground level had a way to simulate a transition function that uses the optimal versions of lower level policies. Our framework is able to simulate a transition function that uses an optimal lower level policy hierarchy and thus can learn multiple levels of policies in parallel.

We introduce a new HRL framework, Hierarchical Actor-Critic (HAC), that can significantly accelerate learning by enabling hierarchical agents to jointly learn a hierarchy of policies. Our framework is primarily comprised of two components: (i) a particular hierarchical architecture and (ii) a method for learning the multiple levels of policies in parallel given sparse rewards.

The hierarchies produced by HAC have a specific architecture consisting of a set of nested, goal-conditioned policies that use the state space as the mechanism for breaking down a task into subtasks. The hierarchy of nested policies works as follows. The highest level policy takes as input the current state and goal state provided by the task and outputs a subgoal state. This state is used as the goal state for the policy at the next level down. The policy at that level takes as input the current state and the goal state provided by the level above and outputs its own subgoal state for the next level below to achieve. This process continues until the lowest level is reached. The lowest level then takes as input the current state and the goal state provided by the level above and outputs a primitive action. Further, each level has a certain number of attempts to achieve its goal state. When the level either runs out of attempts or achieves its goal state, execution at that level ceases and the level above outputs another subgoal.

Figure 1 shows how an ant agent trained with HAC uses its 3-level policy hierarchy $(\pi_2, \pi_1, \pi_0)$ to move through rooms to reach its goal. At the beginning of the episode, the ant's highest level policy, $\pi_2$, takes as input the current state, which in this case is a vector containing the ant's joint positions and velocities $([\theta, \dot{\theta}])$, and its goal state, represented by the yellow box. $\pi_2$ then outputs a subgoal state, represented by the green box, for $\pi_1$ to achieve. $\pi_1$ takes as input the current state and its

goal state represented by the green box and outputs the subgoal state represented by the purple box. Finally, $\pi_0$ takes as input the current state and the goal state represented by purple box and outputs a primitive action, which in this case is a vector of joint torques. $\pi_0$ has a fixed number of attempts to move to the purple box before $\pi_1$ outputs another subgoal state. Similarly, $\pi_1$ has a fixed number of subgoal states that it can output to try to move the agent to the green box before $\pi_2$ outputs another subgoal.

In addition, HAC enables agents to learn multiple policies in parallel using only sparse reward functions as a result of two types of hindsight transitions. *Hindsight action* transitions help agents learn multiple levels of policies simultaneously by training each subgoal policy with respect to a transition function that simulates the optimal lower level policy hierarchy. Hindsight action transitions are implemented by using the subgoal state achieved in hindsight instead of the original subgoal state as the action component in the transition. For instance, when a subgoal level proposes subgoal state $A$, but the next level policy is unsuccessful and the agent ends in state $B$ after a certain number of attempts, the subgoal level receives a transition in which the state $B$ is the action component, not state $A$. The key outcome is that now the action and next state components in the transition are the same, as if the optimal lower level policy hierarchy had been used to achieve subgoal state $B$. Training with respect to a transition function that uses the optimal lower level policy hierarchy is critical to learning multiple policies in parallel, because the subgoal policies can be learned independently of the changing lower level policies. With hindsight action transitions, a subgoal level can focus on learning the sequences of subgoal states that can reach a goal state, while the lower level policies focus on learning the sequences of actions to achieve those subgoal states. The second type of hindsight transition, *hindsight goal* transitions, helps each level learn a goal-conditioned policy in sparse reward tasks by extending the idea of *Hindsight Experience Replay* (Andrychowicz et al. (2017)) to the hierarchical setting. In these transitions, one of the states achieved in hindsight is used as the goal state in the transition instead of the original goal state.

We evaluated our approach on both grid world tasks and more complex simulated robotics environments. For each task, we evaluated agents with 1, 2, and 3 levels of hierarchy. In all tasks, agents using multiple levels of hierarchy substantially outperformed agents that learned a single policy. Further, in all tasks, agents using 3 levels of hierarchy outperformed agents using 2 levels of hierarchy. Indeed, our framework is the first to show empirically that it can jointly learn 3-level hierarchical policies in tasks with continuous state and action spaces. In addition, our approach outperformed another leading HRL algorithm, HIRO (Nachum et al., 2018), on three simulated robotics tasks.

## 2 RELATED WORK

Building agents that can learn hierarchical policies is a longstanding problem in Reinforcement Learning (Sutton et al., 1999; Dietterich, 2000; McGovern & Barto, 2001; Kulkarni et al., 2016; Menache et al., 2002; Şimşek et al., 2005; Bakker & Schmidhuber, 2004; Wiering & Schmidhuber, 1997). However, most HRL approaches either only work in discrete domains, require pre-trained low-level controllers, or need a model of the environment.

There are several other automated HRL techniques that can work in continuous domains. Schmidhuber (1991) proposed a HRL approach that can support multiple levels, as in our method. However, the approach requires that the levels are trained one at a time, beginning with the bottom level, which can slow learning. Konidaris & Barto (2009) proposed Skill Chaining, a 2-level HRL method that incrementally chains options backwards from the end goal state to the start state. Our key advantage relative to Skill Chaining is that our approach can learn the options needed to bring the agent from the start state to the goal state in parallel rather than incrementally. Nachum et al. (2018) proposed HIRO, a 2-level HRL approach that can learn off-policy like our approach and outperforms two other popular HRL techniques used in continuous domains: Option-Critic (Bacon et al. (2017)) and FeUdal Networks (FUN) (Vezhnevets et al. (2017)). HIRO, which was developed simultaneously and independently to our approach, uses the same hierarchical architecture, but does not use either form of hindsight and is therefore not as efficient at learning multiple levels of policies in sparse reward tasks.

## 3 BACKGROUND

We are interested in solving a Markov Decision Process (MDP) augmented with a set of goals $\mathcal{G}$ (each a state or set of states) that we would like an agent to learn. We define an MDP augmented with a set of goals as a *Universal MDP* (UMDP). A UMDP is a tuple $\mathcal{U} = (\mathcal{S}, \mathcal{G}, \mathcal{A}, T, R, \gamma)$, in which $\mathcal{S}$ is the set of states; $\mathcal{G}$ is the set of goals; $\mathcal{A}$ is the set of actions; $T$ is the transition probability function in which $T(s, a, s')$ is the probability of transitioning to state $s'$ when action $a$ is taken in state $s$; $R$ is the reward function; $\gamma$ is the discount rate $\in [0, 1)$. At the beginning of each episode in a UMDP, a goal $g \in \mathcal{G}$ is selected for the entirety of the episode. The solution to a UMDP is a control policy $\pi : \mathcal{S}, \mathcal{G} \rightarrow \mathcal{A}$ that maximizes the value function $v_\pi(s, g) = \mathbb{E}_\pi[\sum_{n=0}^\infty \gamma^n R_{t+n+1} | s_t = s, g_t = g]$ for an initial state $s$ and goal $g$.

In order to implement hierarchical agents in tasks with continuous state and actions spaces, we will use two techniques from the RL literature: (i) the *Universal Value Function Approximator* (UVFA) (Schaul et al., 2015) and (ii) *Hindsight Experience Replay* (Andrychowicz et al., 2017). The UVFA will be used to estimate the action-value function of a goal-conditioned policy $\pi$, $q_\pi(s, g, a) = \mathbb{E}_\pi[\sum_{n=0}^\infty \gamma^n R_{t+n+1} | s_t = s, g_t = g, a_t = a]$. In our experiments, the UVFAs used will be in the form of feedforward neural networks. UVFAs are important for learning goal-conditioned policies because they can potentially generalize Q-values from certain regions of the *(state, goal, action)* tuple space to other regions of the tuple space, which can accelerate learning. However, UVFAs are less helpful in difficult tasks that use sparse reward functions. In these tasks when the sparse reward is rarely achieved, the UVFA will not have large regions of the *(state, goal, action)* tuple space with relatively high Q-values that it can generalize to other regions. For this reason, we also use *Hindsight Experience Replay* (Andrychowicz et al., 2017). HER is a data augmentation technique that can accelerate learning in sparse reward tasks. HER first creates copies of the *[state, action, reward, next state, goal]* transitions that are created in traditional off-policy RL. In the copied transitions, the original goal element is replaced with a state that was actually achieved during the episode, which guarantees that at least one of the HER transitions will contain the sparse reward. These HER transitions in turn help the UVFA learn about regions of the *(state, goal, action)* tuple space that should have relatively high Q-values, which the UVFA can then potentially extrapolate to the other areas of the tuple space that may be more relevant for achieving the current set of goals.

## 4 HIERARCHICAL ACTOR-CRITIC (HAC)

We introduce a HRL framework, Hierarchical Actor-Critic, that can efficiently learn the levels in a multi-level hierarchy in parallel. HAC contains two components: (i) a particular hierarchical architecture and (ii) a method for learning the levels of the hierarchy simultaneously and independently. In this section, we will more formally present our proposed system as a UMDP transformation operation.

The purpose of our framework is to efficiently learn a $k$-level hierarchy $\Pi_{k-1}$ consisting of $k$ individual policies $\pi_0, \ldots, \pi_{k-1}$, in which $k$ is a hyperparameter chosen by the user. In order to learn $\pi_0, \ldots, \pi_{k-1}$ in parallel our framework transforms the original UMDP, $\mathcal{U}_{original} = (\mathcal{S}, \mathcal{G}, \mathcal{A}, T, R, \gamma)$, into a set of $k$ UMDPs $\mathcal{U}_0, \ldots, \mathcal{U}_{k-1}$, in which $\mathcal{U}_i = (\mathcal{S}_i, \mathcal{G}_i, \mathcal{A}_i, T_i, R_i, \gamma_i)$. In the remainder of the section, we will describe these tuples at a high-level. See section 7.3 in the Appendix for the full definition of each UMDP tuple.

### 4.1 STATE, GOAL, AND ACTION SPACES

In our approach, each level of the UMDP hierarchy learns its own deterministic policy: $\pi_i : \mathcal{S}_i, \mathcal{G}_i \rightarrow \mathcal{A}_i, 0 \le i \le k-1$. The state space for every level $i$ is identical to the state space in the original problem: $\mathcal{S}_i = \mathcal{S}$. Since each level will learn to solve a shortest path problem with respect to a goal state, we set the goal space at each level $i$ to be identical to the state space: $\mathcal{G}_i = \mathcal{S}$. Finally, the action space at all levels except the bottom-most level is identical to the goal space of the next level down (i.e. the state space): $\mathcal{A}_i = \mathcal{S}, i > 0$. These levels output subgoal states for the next lower level to achieve. The action space of the bottom-most level is identical to the set of primitive actions that are available to the agent: $\mathcal{A}_0 = \mathcal{A}$.

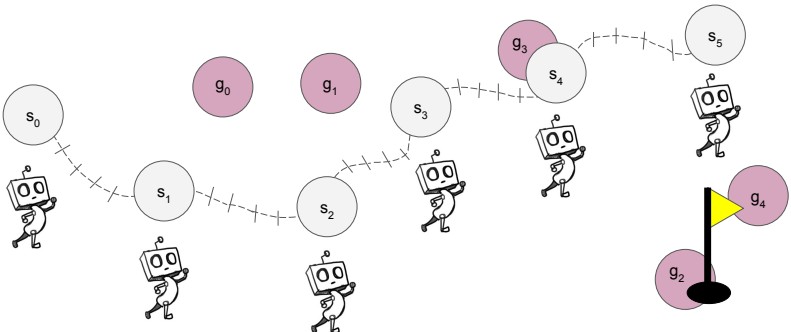

Figure 2: An example episode trajectory for a simple toy example. The tic marks along the trajectory show the next states for the robot after each primitive action is executed. The pink circles show the original subgoal actions. The gray circles show the subgoal states reached in hindsight after at most $H$ actions by the low-level policy.

## 4.2 NESTED POLICIES

HAC learns hierarchies of nested policies. Nesting is critical to decomposing problems because it enables agents to learn tasks requiring long sequences of primitive actions with policies that only need to learn short sequences of actions. HAC nests policies by embedding the policy at level $i-1$, $\pi_{i-1}$, into the transition function at level $i$, $T_i$. The transition function at each subgoal level, $T_i, i > 0$, will work as follows. The subgoal action selected $a_i$ by level $i$ is assigned to be the goal of level $i-1$: $g_{i-1} = a_i$. $\pi_{i-1}$ then has at most $H$ attempts to achieve $g_{i-1}$, in which $H$, or the maximum horizon of a subgoal action, is another parameter provided by the user. When either $\pi_{i-1}$ runs out of $H$ attempts or a goal $g_n, n \geq i-1$, is achieved, the transition function terminates and the agent's current state is returned. Level $i$'s state transition function $T_i$ thus depends on the full policy hierarchy below level $i$, $\Pi_{i-1}$, due to the hierarchy's nested architecture. Each action from $\pi_{i-1}$ depends on $T_{i-1}$, which depends on $\pi_{i-2}$ and so on. Consequently, we use the notation $T_{i|\Pi_{i-1}}$ for level $i$'s state transition function going forward as it depends on the full lower level policy hierarchy. The full state transition function for level $i > 0$ is provided in Algorithm 3 in the Appendix. The base transition function $T_0$ is assumed to be provided by the task: $T_0 = T$.

## 4.3 HINDSIGHT ACTION TRANSITIONS

There are two causes of non-stationary transition functions in our framework that will need to be overcome in order to learn multiple policies in parallel. One cause of non-stationary transition functions is updates to lower level policies. That is, whenever $\pi_i$ changes, the transition function at levels above $i$, $T_{j|\Pi_{j-1}}, j > i$, can change. The second cause is exploring lower level policies. Because all levels have a deterministic policy in our algorithm, all levels will need to explore with some behavior policy $\pi_{i_b}$ that is different than the policy it is learning $\pi_i$. For instance, in continuous domains, the agent may add Gaussian noise to its greedy policy: $\pi_{i_b} = \pi_i + \mathcal{N}(0, \sigma^2)$ for some variance $\sigma^2$. Yet whenever a lower level policy hierarchy uses some behavior policy $\Pi_{i-1_b}$ to achieve a subgoal, the transition function at level $i$, $T_{i|\Pi_{i-1_b}}$, will also vary over time. RL methods will likely not be effective at learning subgoal policies in parallel if each subgoal policy at level $i$ is trained with respect to a transition function that uses the current lower level policy hierarchy $\Pi_{i-1}$ or the behavior lower level policy hierarchy $\Pi_{i-1_b}$. RL methods need the distribution of states to which actions lead to be stable in order to effectively value actions and both $\Pi_{i-1}$ and $\Pi_{i-1_b}$ are continually changing.

In order to overcome these non-stationary issues that hinder the joint learning of policies, HAC instead trains each subgoal policy assuming a transition function that uses the optimal lower level policy hierarchy, $\Pi_{i-1}^*$. $T_{i|\Pi_{i-1}^*}$ is stationary because it is independent of the changing and exploring lower level policies, allowing an agent to learn a policy at level $i$ at the same time the agent learns policies below level $i$.

Hindsight action transitions use a simple technique to simulate the transition function that uses the optimal policy hierarchy below level $i$, $T_{i|\Pi_{i-1}^*}$. In order to explain how hindsight actions transitions are implemented, we will use the example in Figure 2, in which a $k = 2$-level robot is looking to

move from its start state to the yellow flag. The robot begins in state $s_0$ when the high level policy $\pi_1$ outputs the subgoal state $g_0$ for the low level to achieve. The low level policy $\pi_0$ then executes $H = 5$ primitive actions using some behavior policy $\pi_{0_b}$ but is unable to achieve $g_0$, instead landing in $s_1$. After executing $H = 5$ primitive actions, the first action by $\pi_1$ is complete and a hindsight action transition can be created.

Hindsight action transitions have two key components. The first is that the subgoal state achieved in hindsight is used as the action component in the transition, not the originally proposed subgoal state. Thus, the hindsight action transition so far will look like: *[initial state = $s_0$, action = $s_1$, reward = TBD, next state = $s_1$, goal = yellow flag, discount rate = gamma]*. The second key component of the hindsight action transition is the reward function used at all subgoal levels. The first requirement for this reward function is that it should incentivize short paths to the goal because shorter paths can be learned more quickly. The second requirement for the reward function is that it should be independent of the path taken at lower levels. The purpose of hindsight action transitions is to simulate a transition function that uses the optimal lower level policy hierarchy $\Pi_{i-1}^*$. Yet without a model of the environment, the exact path $\Pi_{i-1}^*$ would have taken is unknown. Thus, the reward should only be a function of the state reached in hindsight and the goal state. For each subgoal level, we use the reward function in which a reward of -1 is granted if the goal has not been achieved and a reward of 0 otherwise. Thus, in the example above, the high level of the robot would receive the hindsight action transition *[initial state = $s_0$, action = $s_1$, reward = -1, next state = $s_1$, goal = yellow flag, discount rate = gamma]*, which is the same transition that would have been created had the high level originally proposed state $s_1$ as a subgoal and the transition function used the optimal lower level policy hierarchy to achieve it. Using the same process, the hindsight action transition created for the second action by $\pi_1$ would be *[initial state = $s_1$, action = $s_2$, reward = -1, next state = $s_2$, goal = yellow flag, discount rate = $\gamma$]*.

Although none of the hindsight actions produced in the episode contained the sparse reward of 0, they are still helpful for the high level of the agent. Through these transitions, the high level discovers on its own possible subgoals that fit the time scale of $H$ primitive actions per high level action, which is the time scale that it should be learning. More importantly, these transitions are robust to a changing and exploring lower level policy $\pi_0$ because they assume a transition function that uses $\pi_0^*$ and not the current low level policy $\pi_0$ or low level behavior policy $\pi_{0_b}$.

## 4.4 HINDSIGHT GOAL TRANSITIONS

We supplement all levels of the hierarchy with an additional set of transitions, which we refer to as hindsight goal transitions, that enable each level to learn more effectively in sparse reward tasks by extending the idea of *Hindsight Experience Replay* (Andrychowicz et al., 2017) to the hierarchical setting. As the toy robot example illustrates, it can be difficult for any level in our framework to receive the sparse reward. A level needs to randomly reach its goal state in order to obtain the sparse reward. Hindsight goal transitions use another simple use of hindsight to guarantee that after every sequence of actions by each level in the hierarchy, that level receives a transition containing the sparse reward.

Hindsight goal transitions would be created for each level in the toy robot example as follows. Beginning with the low level, after each of the at most $H = 5$ primitive actions executed by the low level policy $\pi_0$ per high level action, the low level will create two transitions. The first transition is the typical transition non-hierarchical agents create evaluating the primitive action that was taken given the goal state. For instance, assuming the same shortest path reward function described earlier, after the first primitive action in the episode, the low level will receive the transition *[initial state = $s_0$, action = joint torques, reward = -1, next state = first tick mark, goal = $g_0$, discount rate = $\gamma$]*. The second transition is a copy of the first transition, but the goal state and reward components are temporarily erased: *[initial state = $s_0$, action = joint torques, reward = TBD, next state = first tick mark, goal = TBD, discount rate = $\gamma$]*. After the sequence of at most $H = 5$ primitive actions, the hindsight goal transitions will be created by filling in the TBD components in the extra transitions that were created. First, one of the "next state" elements in one of the transitions will be selected as the new goal state replacing the TBD component in each transition. Second, the reward will be updated in each transition to reflect the new goal state. For instance, after the first set of $H = 5$ primitive actions, the state $s_1$ may be chosen as the hindsight goal. The hindsight goal transition created by the fifth primitive action that achieved the hindsight goal would then be *[initial state =*

*4th tick mark, action = joint torques, reward = 0, next state = $s_1$, goal = $s_1$, discount rate = 0].* Moreover, hindsight goal transitions would be created in the same way for the high level of the toy robot, except that the hindsight goal transitions would be made from copies of the hindsight action transitions. Assuming the last state reached $s_5$ is used as the hindsight goal, the first hindsight goal transition for the high level would be *[initial state = $s_0$, action = $s_1$, reward = -1, next state = $s_1$, goal = $s_5$, discount rate = $\gamma$].* The last hindsight goal transition for the high level would be *[initial state = $s_4$, action = $s_5$, reward = 0, next state = $s_5$, goal = $s_5$, discount rate = 0].*

Hindsight goal transitions should significantly help each level learn an effective goal-conditioned policy because it guarantees that after every sequence of actions, at least one transition will be created that contains the sparse reward (in our case a reward and discount rate of 0). These transitions containing the sparse reward will in turn incentivize the UVFA critic function to assign relatively high Q-values to the *(state, action, goal)* tuples described by these transitions. The UVFA can then potentially generalize these high Q-values to the other actions that could help the level solve its tasks.

## 4.5 SUBGOAL TESTING TRANSITIONS

Hindsight action and hindsight goal transitions give agents the potential to learn multiple policies in parallel with only sparse rewards, but some key issues remain. The most serious flaw is that the strategy only enables a level to learn about a restricted set of subgoal states. A level $i$ will only execute in hindsight subgoal actions that can be achieved with at most $H$ actions from level $i - 1$. For instance, when the toy robot is in state $s_2$, it will not be able to achieve a subgoal state on the yellow flag in $H = 5$ primitive actions. As a result, level $i$ in a hierarchical agent will only learn Q-values for subgoal actions that are relatively close to its current state and will ignore the Q-values for all subgoal actions that require more than $H$ actions. This is problematic because the action space for all subgoal levels should be the full state space in order for the framework to be end-to-end. If the action space is the full state space and the Q-function is ignoring large regions of the action space, significant problems will occur if the learned Q-function assigns higher Q-values to distant subgoals that the agent is ignoring than to feasible subgoals that can be achieved with at most $H$ actions from the level below. $\pi_i$ may adjust its policy to output these distant subgoals that have relatively high Q-values. Yet the lower level policy hierarchy $\Pi_{i-1}$ has not been trained to achieve distant subgoals, which may cause the agent to act erratically.

A second, less significant shortcoming is that hindsight action and goal transitions do not incentivize a subgoal level to propose paths to the goal state that the lower levels can actually execute with its current policy hierarchy. Hindsight action and goal transitions purposefully incentivize a subgoal level to ignore the current capabilities of lower level policies and propose the shortest path of subgoals that has been found. But this strategy can be suboptimal because it may cause a subgoal level to prefer a path of subgoals that cannot yet be achieved by the lower level policy hierarchy over subgoal paths that both lead to the goal state and can be achieved by the lower level policy hierarchy.

Our framework addresses the above issues by supplying agents with a third set of transitions, which we will refer to as *subgoal testing* transitions. Subgoal testing transitions essentially play the opposite role of hindsight action transitions. While hindsight actions transitions help a subgoal level learn the value of a subgoal state when lower level policies are optimal, subgoal testing transitions enable a level to understand whether a subgoal state can be achieved by the current set of lower level policies.

Subgoal testing transitions are implemented as follows. After level $i$ proposes a subgoal $a_i$, a certain fraction of the time $\lambda$, the lower level behavior policy hierarchy, $\Pi_{i-1_b}$, used to achieve subgoal $a_i$ must be the current lower level policy hierarchy $\Pi_{i-1}$. That is, instead of a level being able to explore with a noisy policy when trying to achieve its goal, the current lower level policy hierarchy must be followed exactly. Then, if subgoal $a_i$ is not achieved in at most $H$ actions by level $i - 1$, level $i$ will be penalized with a low reward, $penalty$. In our experiments, we set $penalty = -H$, or the negative of the maximum horizon of a subgoal. In addition, we use a discount rate of 0 in these transitions to avoid non-stationary transition function issues.

Using the robot example in Figure 2, after the robot proposes the ambitious subgoal $g_2$ when in state $s_2$, the robot may randomly decide to test that subgoal. The low level policy then has at most $H = 5$ primitive actions to achieve $g_2$. These primitive actions must follow $\pi_0$ exactly. Because the robot

misses its subgoal, it would be penalized with following transition *[initial state = $s_2$, action = $g_2$, reward = -5, next state = $s_3$, goal = Yellow Flag, discount rate = 0]*.

Subgoal testing transitions have three different effects on Q-values depending on the *(state, goal, subgoal action)* tuple that is under consideration. For this analysis, we use the notation $|s - a|$ to refer to the number of actions required by an optimal version of the policy at the level below, $\pi^*_{i-1}$, to move the agent from state $s$ to subgoal state $a$.

1. $|s - a| > H$: For those *(state, goal, subgoal action)* tuples in which the subgoal action could never be completed with $H$ actions by the optimal policy at the level below, the critic function will be incentivized to learn Q-values of $-H$ because the only transitions a subgoal level will receive for these tuples is the penalty transition. Thus, subgoal testing transitions can overcome the major flaw of training only with hindsight action and goal transitions because now the more distant subgoal actions are no longer ignored.

2. $|s - a| \leq H$ and Achievable by $\Pi_{i-1}$: For those *(state, goal, subgoal action)* tuples in which the subgoal action can be achieved by the current lower level policy hierarchy $\Pi_{i-1}$, subgoal testing should have little to no effect. Critic functions will be incentivized to learn Q-values close to the Q-value targets prescribed by the hindsight action and hindsight goal transitions.

3. $|s - a| \leq H$ and Not Achievable by $\Pi_{i-1}$: The effects of subgoal testing are a bit more subtle for those *(state, goal, subgoal action)* tuples in which the subgoal action can be achieved with at most $H$ actions by an optimal version of the policy below, $\pi^*_{i-1}$, but cannot yet be achieved with the current policy $\pi_{i-1}$. For these tuples, critic functions are incentivized to assign a Q-value that is a weighted average of the target Q-values prescribed by the hindsight action/goal transitions and the penalty value of $-H$ prescribed by the subgoal testing transitions. However, it is important to note that for any given tuple there are likely significantly fewer subgoal testing transitions than the total number of hindsight action and goal transitions. Hindsight action transitions are created after every subgoal action, even during subgoal testing, whereas subgoal testing transitions are not created after each subgoal action. Thus, the critic function is likely to assign Q-values closer to the target value prescribed by the hindsight action and hindsight goal transitions than the penalty value of $-H$ prescribed by the subgoal testing transition.

To summarize, subgoal testing transitions can overcome the issues caused by only training with hindsight goal and hindsight action transitions while still enabling all policies of the hierarchy to be learned in parallel. With subgoal testing transitions, critic functions no longer ignore the Q-values of infeasible subgoals. In addition, each subgoal level can still learn simultaneously with lower levels because Q-values are predominately decided by hindsight action and goal transitions, but each level will have a preference for paths of subgoals that can be achieved by the current lower level policy hierarchy.

## 4.6 ALGORITHMS

Algorithm 1 in the Appendix shows the full procedure for Hierarchical Actor-Critic (HAC). Section 7.6 in the Appendix provides additional HAC implementation details. We also provide the discrete version of our algorithm, *Hierarchical Q-Learning* (HierQ), in Algorithm 2 in the Appendix.

## 5 EXPERIMENTS

We evaluated our framework in several discrete state and action and continuous state and action tasks. The discrete tasks consisted of grid world environments. The continuous tasks consisted of the following simulated robotics environments developed in MuJoCo (Todorov et al., 2012): (i) inverted pendulum, (ii) UR5 reacher, (iii) ant reacher, and (iv) ant four rooms. A video showing our experiments is available at `https://www.youtube.com/watch?v=DYcVTveeNK0`. Figure 3 shows some episode sequences from the grid world and inverted pendulum environments for a 2-level agent.

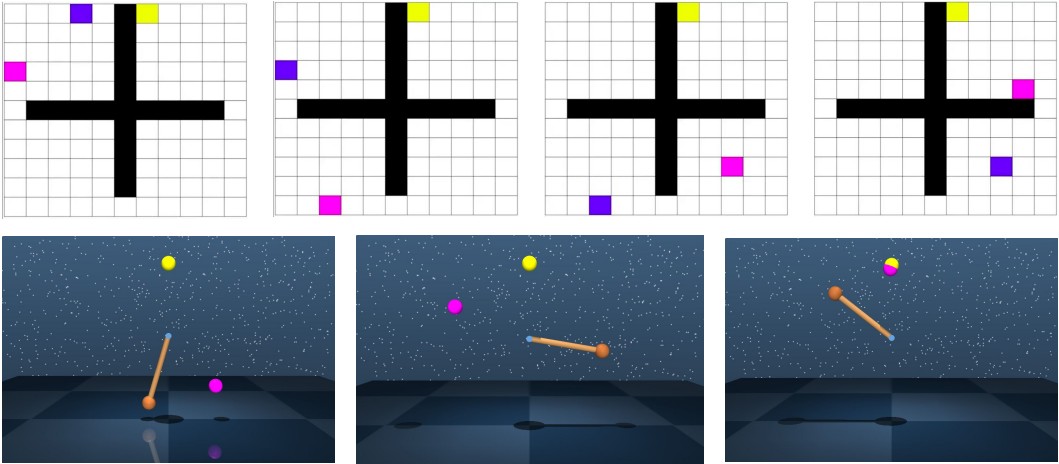

Figure 3: Episode sequences from the four rooms (top) and inverted pendulum tasks (bottom). In the four rooms task, the $k$=2 level agent is the blue square; the goal is the yellow square; the learned subgoal is the purple square. In the inverted pendulum task, the goal is the yellow sphere and the subgoal is the purple sphere.

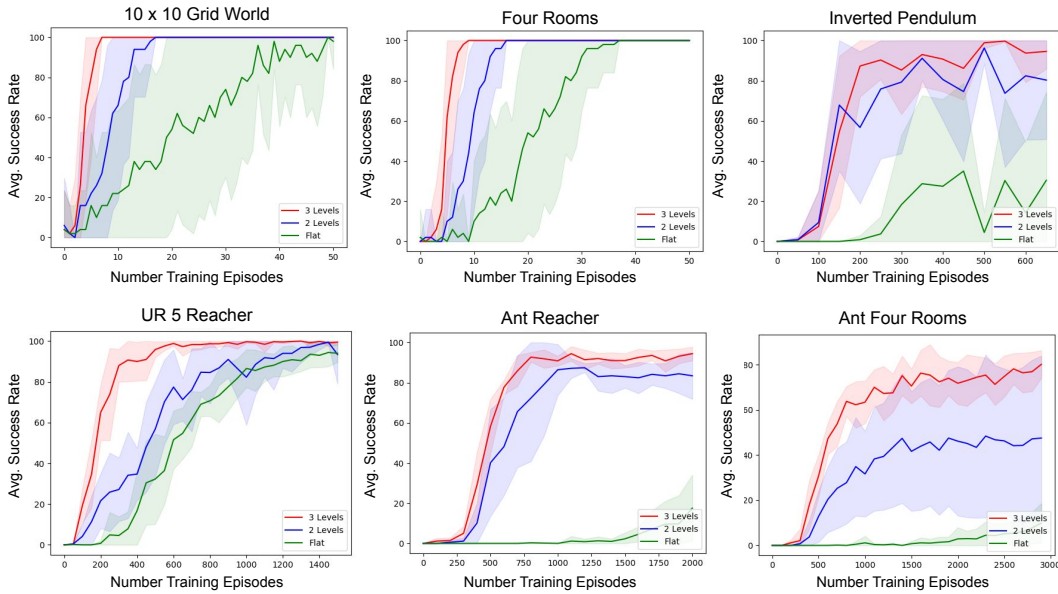

Figure 4: Average success rates for 3-level (red), 2-level agent (blue), and flat (green) agents in each task. The error bars show 1 standard deviation.

## 5.1 RESULTS

We compared the performance of agents using policy hierarchies with 1 (i.e., flat), 2, and 3 levels on each task. The flat agents used Q-learning (Watkins & Dayan, 1992) with HER in the discrete tasks and DDPG (Lillicrap et al., 2015) with HER in the continuous tasks.

Our approach significantly outperformed the flat agent in all tasks. Figure 4 shows the average episode success rate for each type of agent in each task. The discrete tasks average data from 50 trials. The continuous tasks average data from at least 7 trials.

In addition, our empirical results show that our framework can benefit from additional levels of hierarchy likely because our framework can learn multiple levels of policies in parallel. In all tasks, the 3-level agent outperformed the 2-level agent, and the 2-level agent outperformed the flat agent.

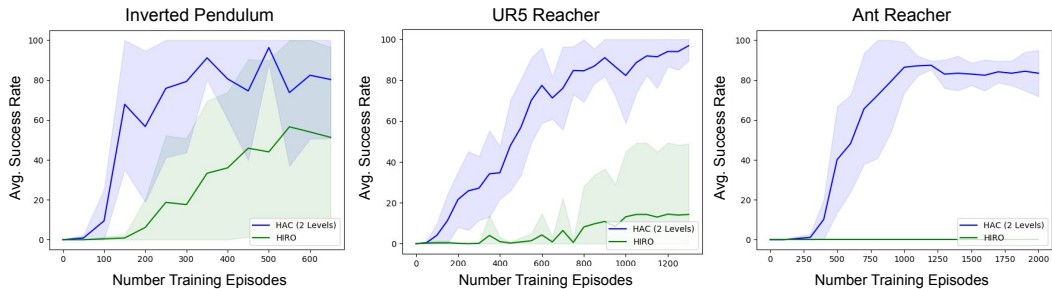

Figure 5: Figure compares the performance of HAC (2 Levels) and HIRO. The charts show the average success rate and 1 standard deviation.

### 5.1.1 BASELINE COMPARISON

We also directly compared our approach HAC to another HRL technique, HIRO (Nachum et al., 2018), which outperforms the other leading HRL techniques that can work in continuous state and action spaces: FeUdal Networks (Vezhnevets et al., 2017) and Option-Critic (Bacon et al., 2017). HIRO enables agents to learn a 2-level hierarchical policy that like our approach can be trained off-policy and uses the state space to decompose a task. Two of the key differences between the algorithms are that (i) HIRO does not use Hindsight Experience Replay at either of the 2 levels and (ii) HIRO uses a different approach for handling the non-stationary transition functions. Instead of replacing the original proposed action with the hindsight action as in our approach, HIRO uses a subgoal action from a set of candidates that when provided to the current level 0 policy would most likely cause the sequence of *(state, action)* tuples that originally occurred at level 0 when the level 0 policy was trying to achieve its original subgoal. In other words, HIRO values subgoal actions with respect to a transition function that essentially uses the current lower level policy hierarchy, not the optimal lower level policy hierarchy as in our approach. Consequently, HIRO may need to wait until the lower level policy converges before the higher level can learn a meaningful policy.

We compared the 2-level version of HAC to HIRO on the inverted pendulum, UR5 reacher, and ant reacher tasks. In all experiments, the 2-level version of HAC significantly outperformed HIRO. The results are shown in Figure 5.

### 5.1.2 SUBGOAL TESTING ABLATION STUDIES

We also implemented some ablation studies examining our subgoal testing procedure. We compared our method to (i) no subgoal testing and (ii) always penalizing missed subgoals even when the lower levels use noisy policies when attempting to achieve a subgoal. Our implementation significantly outperformed both baselines. The results and analysis of the ablation studies are given in section 6 of the Appendix.

## 6 CONCLUSION

Hierarchy has the potential to accelerate learning but in order to realize this potential, hierarchical agents need to be able to learn their multiple levels of policies in parallel. We present a new HRL framework that can efficiently learn multiple levels of policies simultaneously. HAC can overcome the instability issues that arise when agents try to learn to make decisions at multiple time scales because the framework trains each level of the hierarchy as if the lower levels are already optimal. Our results in several discrete and continuous domains, which include the first 3-level agents in tasks with continuous state and action spaces, confirm that HAC can significantly improve sample efficiency.

ACKNOWLEDGEMENTS

This work has been supported in part by the National Science Foundation through IIS-1724237, IIS-1427081, IIS-1724191, and IIS-1724257, NASA through NNX16AC48A and NNX13AQ85G,

ONR through N000141410047, Amazon through an ARA to Platt, Google through a FRA to Platt, and DARPA.

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

# 7 APPENDIX

## 7.1 HIERARCHICAL ACTOR-CRITIC (HAC) ALGORITHM

---

**Algorithm 1** Hierarchical Actor-Critic (HAC)

---

**Input:**
- Key agent parameters: number of levels in hierarchy $k$, maximum subgoal horizon $H$, and subgoal testing frequency $\lambda$.

**Output:**
- $k$ trained actor and critic functions $\pi_0, ..., \pi_{k-1}, Q_0, ..., Q_{k-1}$

**for** $M$ episodes **do**                                          ▷ Train for M episodes
   $s \leftarrow S_{init}, g \leftarrow G_{k-1}$            ▷ Sample initial state and task goal
   $train - level(k-1, s, g)$                             ▷ Begin training
   Update all actor and critic networks
**end for**

**function** TRAIN-LEVEL($i :: level, s :: state, g :: goal$)
   $s_i \leftarrow s, g_i \leftarrow g$                    ▷ Set current state and goal for level $i$
   **for** $H$ attempts or until $g_n, i \leq n < k$ achieved **do**
      $a_i \leftarrow \pi_i(s_i, g_i) + noise$ (if not subgoal testing)    ▷ Sample (noisy) action from policy
      **if** $i > 0$ **then**
         Determine whether to test subgoal $a_i$
         $s_i' \leftarrow train - level(i-1, s_i, a_i)$         ▷ Train level $i-1$ using subgoal $a_i$
      **else**
         Execute primitive action $a_0$ and observe next state $s_0'$
      **end if**
                                                           ▷ Create replay transitions
      **if** $i > 0$ and $a_i$ missed **then**
         **if** $a_i$ was tested **then**                    ▷ Penalize subgoal $a_i$
            $Replay\_Buffer_i \leftarrow [s = s_i, a = a_i, r = Penalty, s' = s_i', g = g_i, \gamma = 0]$
         **end if**
         $a_i \leftarrow s_i'$                           ▷ Replace original action with action executed in hindsight
      **end if**
                                      ▷ Evaluate executed action on current goal and hindsight goals
      $Replay\_Buffer_i \leftarrow [s = s_i, a = a_i, r \in \{-1, 0\}, s' = s_i', g = g_i, \gamma \in \{\gamma, 0\}]$
      $HER\_Storage_i \leftarrow [s = s_i, a = a_i, r = TBD, s' = s_i', g = TBD, \gamma = TBD]$
      $s_i \leftarrow s_i'$
   **end for**
   $Replay\_Buffer_i \leftarrow$ Perform HER using $HER\_Storage_i$ transitions
   **return** $s_i'$                                      ▷ Output current state
**end function**

---

## 7.2 HIERARCHICAL Q-LEARNING (HIERQ) ALGORITHM

HierQ is the version of our algorithm designed for domains with discrete state and action spaces. Note that HierQ does not use subgoal testing. Instead, the algorithm uses pessimistic Q-value initializations to prevent agents from learning to propose subgoal states that are too distant.

---

**Algorithm 2** Hierarchical Q-Learning (HierQ)

---

**Input:**
- Key agent parameters: number of levels in hierarchy $k > 1$, maximum subgoal horizon $H$, learning rate $\alpha$

**Output:**
- $k$ trained Q-tables $Q_0(s, g, a), ..., Q_{k-1}(s, g, a)$

Use pessimistic Q-value initialization: $Q_i(s, g, a) \leq -H^{i+1}$

**for** $M$ episodes **do**                                                  ▷ Train for M episodes
$\quad$ $s_{k-1} \leftarrow S_{init}, g_{k-1} \leftarrow G_{k-1}$               ▷ Sample initial state and task goal
$\quad$                                                      ▷ Initialize previous state arrays for levels $i, 0 < i < k$
$\quad$ $Prev\_States_i \leftarrow Array[H^i]$                          ▷ Length of level $i$ array is $H^i$

$\quad$ **while** $g_{k-1}$ not achieved **do**                             ▷ Begin Training
$\quad\quad$ $a_{k-1} \leftarrow \pi_{k-1_b}(s_{k-1}, g_{k-1})$        ▷ Sample action using $\epsilon$-greedy policy $\pi_{k-1_b}$
$\quad\quad$ $s_{k-1} \leftarrow train - level(k - 2, s_{k-1}, a_{k-1})$    ▷ Train next level
$\quad$ **end while**
**end for**

**function** TRAIN-LEVEL($i :: level, s :: state, g :: goal$)
$\quad$ $s_i \leftarrow s, g_i \leftarrow g$                               ▷ Set current state and goal for level $i$
$\quad$ **for** $H$ attempts or until $g_n, i \leq n < k$ achieved **do**
$\quad\quad$ $a_i \leftarrow \pi_{i_b}(s_i, g_i)$                        ▷ Sample action using $\epsilon$-greedy policy $\pi_{i_b}$
$\quad\quad$ **if** $i > 0$ **then**
$\quad\quad\quad$ $s_i^{'} \leftarrow train - level(i - 1, s_i, a_i)$        ▷ Train level $i - 1$ using subgoal $a_i$
$\quad\quad$ **else**
$\quad\quad\quad$ Execute primitive action $a_0$ and observe next state $s_0^{'}$
$\quad\quad\quad\quad$                              ▷ Update $Q_0(s, g, a)$ table for all possible subgoal states
$\quad\quad\quad$ **for** each state $s_{goal} \in S$ **do**
$\quad\quad\quad\quad$ $Q_0(s_0, s_{goal}, a_0) \leftarrow (1-\alpha) \cdot Q_0(s_0, s_{goal}, a_0) + \alpha \cdot [R_0 + \gamma max_a Q_0(s_0^{'}, s_{goal}, a_0)]$
$\quad\quad\quad$ **end for**
$\quad\quad\quad$                              ▷ Add state $s_0$ to all previous state arrays
$\quad\quad\quad$ $Prev\_States_i \leftarrow s_0, 0 < i < k$
$\quad\quad\quad$                              ▷ Update $Q_i(s, g, a), 0 < i < k$, tables
$\quad\quad\quad$ **for** each level $i, 0 < i < k$ **do**
$\quad\quad\quad\quad$ **for** each state $s \in Prev\_States_i$ **do**
$\quad\quad\quad\quad\quad$ **for** each goal $s_{goal} \in S$ **do**
$\quad\quad\quad\quad\quad\quad$ $Q_i(s, s_{goal}, s_0^{'}) \leftarrow (1 - \alpha) \cdot Q_i(s, s_{goal}, s_0^{'}) + \alpha \cdot [R_i + \gamma max_a Q_i(s_0^{'}, s_{goal}, a)]$
$\quad\quad\quad\quad\quad$ **end for**
$\quad\quad\quad\quad$ **end for**
$\quad\quad\quad$ **end for**
$\quad\quad$ **end if**
$\quad\quad$ $s_i \leftarrow s_i^{'}$
$\quad$ **end for**
$\quad$ **return** $s_i^{'}$                                                  ▷ Output current state
**end function**

---

## 7.3 UMDP TUPLE DEFINITIONS

We now formally define the UMDPs tuples for all levels.

$\mathcal{U}_0$: This is the lowest level of the hierarchy. It has the same state set, action set, and state transition function as $\mathcal{U}_{original}$: $\mathcal{S}_0 = \mathcal{S}, \mathcal{A}_0 = \mathcal{A}$, and $T_0 = T$. The goal states for which $\mathcal{U}_0$ will be responsible for learning will be dictated by the UMDP one level higher, $\mathcal{U}_1$. However, given that every state is potentially a goal, the goal space is defined to be the state space: $\mathcal{G}_0 = \mathcal{S}$. The framework is flexible as to the reward function used at level 0, but we will use the shortest path reward function, in which a reward of 0 is granted if the next state, $s'$, maps to the goal and -1 otherwise: $R_0(s', g) = 0, \forall s' : s' \in g$ and $R_0(s', g) = -1, \forall s' : s' \notin g$. In addition, the discount rate is set to 0 if the goal has been achieved, but remains $\gamma$ otherwise: $\gamma_0(s', g) = 0, \forall s' : s' \in g$, $\gamma_0(s', g) = \gamma, \forall s' : s' \notin g$. The objective in $\mathcal{U}_0$ is to find the policy $\pi_0 : \mathcal{S}_0, \mathcal{G}_0 \to \mathcal{A}_0$ that maximizes the value function $v_{\pi_0}(s, g) = \mathbb{E}_{\pi_0}[\sum_{n=0}^{\infty} \gamma_0^n R_{0_{t+n+1}} | s_{0_t} = s, g_{0_t} = g]$ for the initial state and goal combinations that the policy $\pi_1$ from $\mathcal{U}_1$ requires.

$\mathcal{U}_i, 1 \leq i < k$: These UMDPs represent the remainder of the hierarchy. The state space is the same as the state space in $U_{original}$: $\mathcal{S}_i = \mathcal{S}$. For all levels except for the top level, the set of goals is dictated by the level above. These goals can potentially be any state: $G_i = \mathcal{S}, 1 \leq i < k-1$. For the top level, the goal set is provided by the task: $\mathcal{G}_{k-1} = \mathcal{G}$. The action space is the state space as these policies will output subgoal states for the next level to achieve: $\mathcal{A}_i = \mathcal{S}$. The state transition function consists of two transition functions: $T_{i|\hat{\Pi}_{i-1}}(s, a) = \{$(i) $T_{i|\Pi_{i-1}^*}(s, \bar{a}), \bar{a} = T_{i|\Pi_{i-1_b}}(s, \pi_{i_b}(s, g_i))$, (ii) $T_{i|\Pi_{i-1}}(s, a)\}$. The first transition function will be used to generate hindsight transitions. Note that the action $\bar{a}$ passed to the first transition function must be generated by some hierarchical behavior policy $\Pi_{i_b}$. The second transition function will be used for subgoal testing and will follow the current lower level policy hierarchy exactly. The reward function is $R_i(a, s', g) = penalty, \forall s' : s' \notin a$, $R_i(a, s', g) = -1, \forall s' : (s' \in a) \wedge (s' \notin g)$, and $R_i(a, s', g) = 0, \forall s' : (s' \in a) \wedge (s' \in g)$. The penalty reward is only issued during subgoal testing. $\gamma_i$ is set to 0 if a subgoal is tested and missed or if an action achieves the goal, but is otherwise $\gamma$ from $\mathcal{U}_{original} : \gamma_i(a, s', g) = 0, \forall s' : (s' \notin a) \vee (s' \in g)$. The objective in each $U_i$ is to learn a policy $\pi_i : \mathcal{S}_i, \mathcal{G}_i \to \mathcal{A}_i$ that maximizes the value function $v_{\pi_i|\hat{\Pi}_{i-1}}(s, g) = \mathbb{E}_{\pi_i|\hat{\Pi}_{i-1}}[\sum_{n=0}^{\infty} \gamma^n R_{i_{t+n+1}} | s_{i_t} = s, g_{i_t} = g]$.

## 7.4 UMDP STATE TRANSITION FUNCTION

---

**Algorithm 3** UMDP $\mathcal{U}_i$ Transition Function: $T_{i|\Pi_{i-1}}(state, action)$

---

**Input:** state $s$, action $a$,
**Output:** next state

**return** $Execute - H - Actions(s, a, i - 1, H)$      ▷ Execute $\leq H$ actions using policy $\pi_{i-1}$

**function** EXECUTE-H-ACTIONS($s :: state, a :: action, i :: level, itr :: iteration$)
     $s' = T_{i|\Pi_{i-1}}(s, \pi_i(s, a))$      ▷ Execute 1 action using policy $\pi_i$
     itr -= 1      ▷ Decrement iteration counter
     **if** $itr == 0$ or $s' \in g, \forall g \in \{g_i, ..., g_{k-1}\}$ **then**
         **return** $s'$      ▷ Return next state if out of iterations or goal achieved
     **else**
         **return** $Execute - H - Actions(s', a, i, itr)$      ▷ Execute another action from $\pi_i$
     **end if**
**end function**

---

## 7.5 SUBGOAL TESTING ABLATION STUDIES

Both the qualitative and quantitative results of the subgoal testing ablation studies support our implementation. When no subgoal testing was used, the results were as expected. The subgoal policies would always learn to set unrealistic subgoals that could not be achieved within $H$ actions by the level below. This led to certain levels of the hierarchy needing to learn very long sequences of actions that the level was not trained to do. When the Q-values of these unrealistic subgoal states were examined, they were high, likely because there were no transitions indicating that these should have low Q-values.

The implementation of always penalizing subgoals even when a noisy lower level policy hierarchy was used also performed significantly worse than our implementation. One likely reason for this

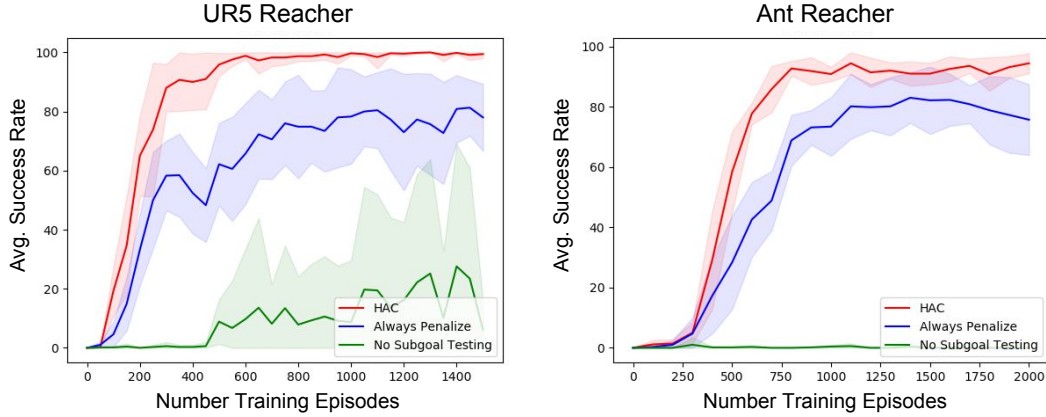

Figure 6: Results from the ablation studies examining our subgoal testing procedure. We compare our implementation to two other options: (i) no subgoal testing and (ii) an implementation in which all missed subgoals are penalized even when lower levels use noisy policies to try to achieve the subgoal state that is being tested. 3-level agents were used in all options.

outcome is that always penalizing strategy incentivizes subgoal levels to output overly conservative subgoals, which means a subgoal level needs to learn longer sequences of subgoals that in turn take longer to learn. Subgoal levels are incentivized to set nearby subgoals because more distant ones are less likely to be achieved when noise is added to actions.

## 7.6 HIERARCHICAL ACTOR-CRITIC IMPLEMENTATION DETAILS

Below we provide some of the important details of our implementation of HAC. For further detail, see the GitHub repository available at `https://github.com/andrew-j-levy/Hierarchical-Actor-Critc-HAC-`.

- DDPG Parameters:
  - Bounded Q-Values: We bound the output of each critic function to the range $[-H, 0]$ using a negative sigmoid function. An upper bound of 0 was chosen because all rewards in HAC are nonpositive so positive Q-values are illegal. We believe the lower bound of $-H$ is helpful for learning Q-values as the critic function does not need to learn precise Q-values for the large space of irrelevant actions in which the current stat is far from the goal state. The bound of $-H$ also ensures that subgoal states that were reached in hindsight should have higher Q-values than any subgoal state that is too distant and penalized during subgoal testing.
  - DDPG Target Networks: For the 2- and 3-level agents, we removed the target networks used in DDPG because we saw in some of our early MuJoCo experiments that these hierarchical agents performed at least as well without target networks. We evaluated the flat agents with and without target networks. The flat agent generally performed better with target networks, but our results show the better result of the two.
  - Exploration: Each level uses the following exploration strategy when a level is not involved in subgoal testing.
    * 20% of actions are sampled uniformly at random from the level's action space
    * 80% of actions are the sum of actions sampled from the level's policy and Gaussian noise
  - Neural Network Architectures: All actor and critic neural networks had 3 hidden layers, with 64 nodes in each hidden layer. ReLU activation functions were used.
- HAC Parameters:
  - Maximum horizon of a subgoal, $H$:
    1. For $k$=3-level agents in MuJoCo tasks, $H = 10$

2. For $k$=2-level agents in MuJoCo tasks, $H$ was generally in the range [20,30]

– Subgoal testing rate $\lambda = 0.3$
– Goal and subgoal achievement thresholds were hand-crafted.

