# OpenReview forum: "Learning Multi-Level Hierarchies with Hindsight"
_ICLR.cc/2019/Conference_

### Official Review · AnonReviewer1 · 2018-11-01
**An HRL framework with an arbitrary number of levels**

**Rating:** 5
**Confidence:** 4

**Review:**

This paper proposed a framework that can improve the performances of reinforcement learning algorithms in tasks that involve long time horizons and sparse rewards. The proposed method is a hierarchical reinforcement learning framework that can use policy hierarchies with an arbitrary number of levels. To improve the sample efficiency in the learning process, the authors proposed to apply the hindsight experience replay mechanism at each level. Also, in order to avoid the actor function to output an unrealistic subgoal, the authors proposed the subgoal testing technique.

The proposed framework is interesting. And the example in Section 3.5 clearly demonstrate how this framework works. The authors proposed to solve a UMDP by solving a hierarchy of k UMDPs, where k is a hyperparameter. Each level (except for the bottom most level) will output subgoal states for the next level to achieve. This hierarchy is reasonable and easy to understand. However, from the definition on Page 3, it seems that all of the intermediate levels i (the case where 0 < i < k - 1) has the same state and action spaces. They are all equal to the state set of the original UMDP. Under this setting, will adding more intermediate levels help improve the performance a lot? We only see results with at most one intermediate level in the experiment. It will be better if the authors can show results on more levels (i.e. at least 4 levels in total).

Moreover, the proposed framework has a policy limit parameter T, meaning that we only consider if a goal can be achieved within T steps or not, at each level. Is this parameter necessary to be the same for all levels? Also, it will be better if the author can show some results on the performances of the proposed method according to different values for T. The authors also proposed the subgoal testing technique. It is also better if the authors can show some performance comparisons on the cases with and without this technique.

The authors claimed that their method has the advantage over some existing HRL methods (e.g. the Option-Critic Architecture [1]) that their method can use policy hierarchies with an arbitrary number of levels while these methods can only use policy hierarchies with two levels. In the experiments, the authors also showed that, in some of their experiments, the 3-layer agent (with 2 subgoal layers) outperforms the 2-layer agent (with 1 subgoal layer), under their framework. However, the authors did not compare their 2-layer agent's performance with these existing HRL methods, which means that we do not know if their 3-layer agent's performance is better than that of some of the existing 2-layer agent methods. In addition to that, as I mentioned before, it is better if the authors can show experiment results on more levels (e.g. 4 levels and more) to show that their method can perform well in practice for policy hierarchies with many levels.


References:

[1] Pierre-Luc Bacon, Jean Harb, and Doina Precup. The option-critic architecture. CoRR, abs/1609.05140, 2016.

---

### Official Review · AnonReviewer3 · 2018-11-02
**Very nice approach for hierarchical RL**

**Rating:** 7
**Confidence:** 4

**Review:**

This paper presents a novel approach for doing hierarchical deep RL. Each level of the hierarchy is rewarded for reaching a goal state. The top level's goal state is the environment goal, lower level goals states are the actions of the higher levels. The lowest level's actions are primitive actions. Each level can act until it reaches it goal or a maximum of T steps. Then HER is used to still learn from missed subgoals. For example, if the lowest level is given a subgoal and fails to achieve it, it is trained with a new experience where the goal was the achieved state. In addition, the level above is trained with an experience where the action it chose (the subgoal that was not achieved) is replaced with the subgoal that was achieved. So HER is replacing goals on one level and replacing actions on the higher level. The paper shows nice empirical results across 6 domains.

The two main differences from prior work are:
1. Explicit constraint on how long the policies at each level can be.
2. Use of HER in a novel way (on goals and actions) to learn from failed attempts at reaching subgoals from lower levels.

The use of HER in this work is really powerful and everything fits together nicely to make it work.

The only un-satisfying part of the algorithm is the need for a subgoal testing phase. Some actions are randomly decided to be testing phases, where all exploration is turned off at lower levels and the agent at the level selecting that subgoal is given negative reward if the subgoal is not achieved. This feels a bit unnatural to me. Does it not work if you punish a level for selecting a failed subgoal even if exploration is on? Does this phase unnecessarily punish levels for selecting subgoals that aren't reached early in learning, where even with no exploration a lower level may not have learned to reach the subgoal yet?

The main drawback of the paper is that there is no empirical comparison to related work. Instead the approach is only compared to doing learning with no hierarchy. Still, in all 6 domains, there is a clear improvement to using the hierarchy vs a flat hierarchy.

Pros:
- Nice approach for hierarchical deep RL
- Great use of HER to improve subgoal learning
- Good empirical results showing benefit of approach over flat learning
Cons:
- No empirical comparison to related work
- Subgoal testing phase seems a bit hacky.

---

### Official Review · AnonReviewer2 · 2018-11-03
**The technique is sound and demonstrated good performance on a range of RL tasks, however its significance is not fully demonstrated.**

**Rating:** 6
**Confidence:** 4

**Review:**

Pros:
1. A nice idea combining universal MDP formulation and Hindsight experience replay for HRL that can deal with hierarchies with more than two levels of policies in continuous tasks.
2. Good empirical results

Cons:
1. One limitation of this work is that the goal set is known. What if the goals are unknown?

2. The current domains seems relative simple comparing other existing papers on HRL, hence it is hard to tell the significance of the method.

3. It Lacks thorough experimental analysis. Some comments are suggestions are provided here.
---Since the proposed framework can deal with arbitrary level of hierarchies, it might be better to include include an experiment comparing the more than 2 subgoal layers. This will help understand whether there is any diminishing return by increasing the number of layers.

---What kind of policy representations and hyperparameters of the training algorithm are used? Are they the same for different domains? Some critical details and some ablation test should be provided.

---The paper can also be strengthened if some comparisons to other HRL methods can be included.

---

### Comment · Area_Chair1 · 2018-11-19
**mixed reviews;  author response?  further input from reviewers?**

Thank you to the assigned reviewers and the anonymous reviewer for your feedback.
It would be good to hear a response from the authors. Similarly, the reviewers can also provide additional comments after having read the other reviews.  Now is the time for further discussion.
Open issues include: (i) comparison to other HRL methods and claims of originality; (ii) what is the impact of having >2 levels?

Area chair

---

> ### Comment · AnonReviewer3 · 2018-11-27
> **Revisions and related work**
>
> The added comparison to the HIRO in the related work and experiment sections is nice, and addresses my main concern with the paper. I'm maintaining my score of 7 - Accept.

---

### Author Response · Authors · 2018-11-21
**Hierarchical RL Algorithm Comparison Implemented (1 of 2)**

Dear Reviewers, Anonymous Reviewer, and Area Chair,

Thank you for your helpful feedback.  In this note, we would like to provide an update on the hierarchical RL comparison we have implemented.  We will post a separate comment shortly addressing the other questions and concerns the reviewers and anonymous reviewer had.

We have completed a comparison to the algorithm HIRO (HIerarchical Reinforcement learning with Off-policy correction)[1].  We chose to compare to HIRO because (i) the algorithm has been peer-reviewed as it was recently accepted to the NeurIPS (NIPS) 2018 conference and (ii) the method has proven that it can outperform the other leading hierarchical RL algorithms that work in continuous state and action spaces: FuN (FeUdal Networks)[2] and Option-Critic[3].  The paper introducing HIRO is available at https://arxiv.org/abs/1805.08296.

We compared the two-level version of our approach, Hierarchical Actor-Critic (HAC), to HIRO, which by design has two levels, on both the relatively easy inverted pendulum task and the relatively more difficult UR5 reacher task.  In both tasks, the two-level version of HAC outperforms HIRO.  The outperformance is particularly substantial in the UR5 reacher task as HIRO is unable to maintain a goal achievement success rate > 0%, while the two-level version of HAC can achieve a 90+% success rate in around 900 episodes.  We will provide these performance comparison charts in the revised paper.

Below we discuss two key differences between the algorithms that we believe enable HAC to outperform HIRO.  We will explain the remaining key differences in the revised paper.

The first significant difference between the algorithms is that HIRO does not use Hindsight Experience Replay (HER)[4] at the upper level of the hierarchy.  HIRO also does not use HER at the lower level and instead uses a distance-based reward function.  The data augmentation technique HER is critical because it helps the upper-level Universal Value Function Approximator (UVFA)[5] learn about the (state, action, goal) tuples that should have high Q-values.  In other words, HER helps the higher-level UVFA learn about the helpful subgoal actions that can move the agent from the current state to goals throughout the goal space.  This is important because the UVFA can then attempt to transfer these high Q-values to the subgoal actions that are relevant for achieving the higher level’s current set of task goals.  HER is particularly important for relatively difficult tasks, such as UR5 Reacher, in which it is unlikely that the goal can be achieved with a random policy.  Without HER in these situations, because the sparse reward is rarely achieved and because there are no high Q-values to transfer, the UVFA will likely be unable to assign high Q-values to the subgoal actions that can solve the task, which should slow down learning.

---

### Author Response · Authors · 2018-11-21
**Hierarchical RL Algorithm Comparison Implemented (2 of 2)**

The second key difference is how each algorithm handles the non-stationarity of the higher-level state transition functions.  Consider the situation in which a two-level agent is in state A with a task goal of state C.  The higher-level proposes state B as a subgoal, but in less than T low-level actions, the agent does end up achieving the goal state C.  Due to its strategy of replacing the original subgoal action with the subgoal state that was achieved, the higher level within the HAC agent would receive the transition [state = A, subgoal action = C, reward = 0, next state = C, goal = C].  In other words, the upper level within HAC would replay the episode as if it had originally proposed state C as the subgoal.  On the other hand, HIRO’s off-policy correction strategy chooses the action component in the higher level transition to be the subgoal state that would most likely cause the sequence of (state, action) tuples that occurred at the lower level when the low-level policy was trying to achieve the original subgoal state B.  In the example above, this will most likely be state B or some other state that is not state C.  Thus, the higher level HIRO agent will likely receive the transition [state = A, subgoal action = B, reward = cumulative low level reward, next state = C, goal = C].  We believe that using state B or some other state that is not state C as the subgoal action component is a critical error because this transition likely will not be valid in the future, while the transition passed to HAC likely will.  As the lower level continues to improve, at some point proposing subgoal state B will not result in the agent ending in state C.  This renders the update HIRO made obsolete.  At the same time, the lower level policy should be able to learn to move the agent from state A to at least close to state C.  When this occurs and HIRO again needs to decide on which subgoal state action to choose for the above high-level transition, it is likely that it will choose state C as the subgoal that would most likely cause the low-level (state, action) tuples that previously occurred given the current low level policy.  The higher level in HIRO would then make a similar update to the one that the higher level within HAC made possibly much earlier in training.

The main result here is that because HAC agents replace the original subgoal with the actual subgoal state achieved, HAC agents can learn all policies within the hierarchy in parallel.  On the other hand, HIRO may need to wait for the policy at one level to converge before it can accurately train the policy at the next higher level, which should cause HIRO to learn more slowly than HAC.  This also has important consequences for adding more levels to the hierarchy.  Because HAC learns all policies in parallel, adding more levels can be helpful as it can shorten the sequence of actions that each level needs to learn.  However, when one policy is learned at a time as in HIRO, there is less of a benefit to inserting additional levels into the hierarchy.


[1]  O. Nachum, S. Gu, H. Lee, and S. Levine, “Data-efficient hierarchical reinforcement learning,” CoRR , vol. abs/1805.08296, 2018.

[2]  Alexander Sasha Vezhnevets, Simon Osindero, Tom Schaul, Nicolas Heess, Max Jaderberg, David Silver, and Koray Kavukcuoglu. Feudal networks for hierarchical reinforcement learning. CoRR, abs/1703.01161, 2017. URL http://arxiv.org/abs/1703.01161.

[3]  Pierre-Luc Bacon, Jean Harb, and Doina Precup. The option-critic architecture. CoRR ,
abs/1609.05140, 2016. URL http://arxiv.org/abs/1609.05140 .


[4]  M. Andrychowicz, F. Wolski, A. Ray, J. Schneider, R. Fong, P. Welinder, B. McGrew, J. Tobin, P. Abbeel, and W. Zaremba, “Hindsight experience replay,” in NIPS , 2017.

[5]  Tom Schaul, Daniel Horgan, Karol Gregor, and David Silver. Universal value function approximators. In Francis Bach and David Blei (eds.), Proceedings of the 32nd International Conference on Machine Learning , volume 37 of Proceedings of Machine Learning Research , pp. 1312–1320, Lille, France, 07–09 Jul 2015. PMLR. URL http://proceedings.mlr.press/v37/
schaul15.html .

---

### Author Response · Authors · 2018-11-21
**Comparison results added to revised paper**

We have added the graphs containing our comparison results to the revised paper (see page 9).  Please note that we will be making further revisions to the paper.

---

### Author Response · Authors · 2018-12-02
**Response to Remaining Reviewer Concerns**

Dear Reviewers and Area Chair,

In the post below, we address many of the Reviewers’ remaining concerns.

R1/R3 - How does Hierarchical Actor-Critic (HAC) perform with 4-level agents?

We have not run any experiments with 4-level agents using the latest iteration of HAC.  However, we did previously run an experiment with 4-level agents in the inverted pendulum task using a slightly different version of HAC.  In this older version of HAC, the subgoal space was the concatenation of the desired Cartesian (x,y,z) end-effector position and the desired angular velocity of the pendulum (i.e. a 4-dim subgoal space).  Whereas in the latest version of HAC, the subgoal space in the inverted pendulum task is just the state space, which is the 2-dim vector including the angle of pendulum and the angular velocity of the pendulum.  A video of the 4-level agents in the inverted pendulum task is available at https://www.youtube.com/watch?v=Q_NGMkQ29oU .  With the older version of HAC, we did find that the 4-level agents outperformed agents using 1, 2, and 3 levels of hierarchy in the inverted pendulum task.


R2/R3 - Can you provide ablation tests examining subgoal testing?  (R3) How does HAC perform without subgoal testing?  (R2) Does exploration noise need to be turned off when subgoal testing?

We have not implemented any ablation tests that examine subgoal testing using the latest version of HAC.  However, we did perform limited ablation tests using previous versions of the algorithm, and the results do support our implementation of subgoal testing.  With these previous versions of the algorithm, agents that did not use subgoal testing were not able to solve the UR5 Reacher task.  Similarly, we did previously implement an agent in the inverted pendulum task that penalized all missed subgoals.  In the single trial that was implemented, performance was significantly worse than when noise was turned off during subgoal testing.  This latter result may have occurred because penalizing all missed subgoals even when exploration noise is added may disincentivize a level from setting distant subgoals as these are more likely to be missed when a level uses a noisy policy.  When a level needs to set closer subgoals, the level needs to learn a longer sequence of subgoal actions, which can slow learning.


R3 - Is it necessary to use the same value for the policy limit parameter H (“T” in initial draft) for each level?

No, the policy limit parameter for each level can be set to whichever value the user prefers.  We designed HAC to use a single value for H for two reasons.  First, using different values of H for each level may hurt the ability of agents to equitably divide the task amongst its levels.  Different values for H will mean that one level needs to learn a longer sequence than the others, and learning may slow if this difference is large.  Second, a single value for all levels limits the number of hyperparameters that need to be set by the user.


R1 - What if the goals are unknown?

We assume the scenario in which the agent is given a set of goal states to learn to achieve.  However, this set of goal states can include all possible goal states so the designer is not required to specify a particular set of goals.

---

### Meta-Review · Area_Chair1 · 2018-12-15
**lean towards accept**

**Confidence:** 4
**Recommendation:** Accept (Poster)

**Metareview:**

As per R3: This paper presents a novel approach for doing hierarchical deep RL (HRL) on UMDPs by:
(a) use of hindsight experience replay at multiple levels;  combined with (b) max T timesteps
at each level. By effectively learning from missed goals at multiple levels, it allows for fairly
data-efficient learning and can (in principle) work for an arbitrary number of levels.
HRL is an important open problem.

The weaknesses described reviewers include limited comparisons to other HRL methods; its applicaiton to fairly simple domain;
its still unclear what the benefit of >=4 levels is, and what the diminishing returns are wrt to the claim of working
for an arbitrary number of levels.  R1(5) and R3(7)  stand by their scores. R1(5) still has some remaining concerns
regarding some experiments not being done across all tasks, an older version of the HAL algo baseline being used, and
lack of insight regarding >= 4 levels.

Based on the balance of the reviewers comments and the AC's own reading of the paper and results,
and the importance of the problem, the AC leans towards accept.  Using Hindsight Exp Replay across multiple levels
is a simple-but-interesting idea, and the terminate-after-T steps is an interesting heuristic to make this effective.
While the paper does not give insight for large (>=4) levels, it does make for an interesting framework that
will inspire further work.  The AC recommends that the claims regarding an "arbitrary number of levels" be significantly toned down.